# Factors Influencing Treatment Outcome and Proprioception after Electrocoagulation of the Femoral Insertion of the Anterior Cruciate Ligament

**DOI:** 10.3390/ijerph192013569

**Published:** 2022-10-20

**Authors:** Adam Pogorzała, Ewa Kądzielawska, Łukasz Kubaszewski, Mikołaj Dąbrowski

**Affiliations:** 1Institute of Applied Mechanics, Poznan University of Technology, 60-965 Poznan, Poland; 2Adult Spine Orthopaedics Department, Poznan University of Medical Sciences, 61-545 Poznan, Poland

**Keywords:** ACL, injury, vaporization, electrocoagulation, thermal shrinkage, proprioception, exercises, dominant limb

## Abstract

(1) Background: Studies have established that exercises shaping the sense of deep sensation are an important element of medical rehabilitation of patients after vaporization of the femoral insertion of the anterior cruciate ligament and affect the restoration of correct movement patterns, thus reducing the risk of injuries. The aim of this study was to determine the factors influencing the treatment outcome and deep-feeling function after applying a specific rehabilitation scheme 12 weeks after anterior cruciate ligament electrocoagulation surgery. (2) Methods: The study group consisted of 41 patients after partial rupture of the anterior cruciate ligament, who underwent electrocoagulation of the femoral cruciate ligament attachment and microfracture of the femoral attachment area. All patients were operated on by the same surgeon and then rehabilitated according to the same medical rehabilitation protocol. The anthropometric and clinical data were collected through an anterior drawer test, Lachman test, assessment of the range of movements in the knee joint, muscle strength test, Unterberger test and Lysholm questionnaire. The assessment was performed before the surgery, and then on days 7–10, after 6 and 12 weeks of rehabilitation treatment. (3) Results: Statistical improvement of the parameters was demonstrated by strength of the quadriceps and hamstrings muscle, the Unterberger test, and the Lysholm scale after surgery. A negative correlation was found between the Unterberger test and Lysholm scale at the end of the research period and it differed depending on the gender and the dominant limb. The Lysholm scale and muscle strength were independent of sex, dominant extremity and associated damage of the meniscus and cartilage. The Lysholm scale 6 weeks after surgery negatively correlated with BMI. (4) Conclusions: Stability of the knee joint and improvement of proprioception were demonstrated 12 weeks after treatment with an ACL electrocoagulation and rehabilitation regimen. The factors contributing to a better treatment outcome were greater muscle strength, less thigh asymmetry, better sense of depth, younger age and lower body weight.

## 1. Introduction

According to epidemiological data, anterior cruciate ligament (ACL) injuries affect 68.6 out of 100,000 people, with a particular emphasis on people actively practicing sports [1]. In the case of partial isolated ACL injuries, the electrocoagulation (electrocoagulation, thermal shrinkage) of the femoral attachment may be considered [2]. Surgical treatment should always be supplemented with properly selected and conducted medical rehabilitation, which should recreate the range of motion in the knee joint and strength of the thigh muscles, as well as the sense of deep feeling [3]. The above activities are intended to regain the patient’s fitness from before the injury, and also constitute a preventive measure against repeated injuries [4]. The risk of re-damage to ligament structures at different times of ligament healing may vary. A physiotherapist who is planning a rehabilitation program should be well-versed in the risks associated with the ligamentization process of the ligament and, if necessary, modify the rehabilitation program on an ongoing basis.

The cause of the imbalance may be peripheral damage (e.g., ligament injuries, joint dislocations, fractures, etc.) and the clinical symptom may be, among other things, a feeling of joint instability. In addition to causal treatment, it recommends the use of medical rehabilitation methods, the task of which is to restore the sense of balance, for example, through the use of deep sensation exercises.

There were differences in the results of surgical treatment of ACL reconstruction depending on age, sex, dominant limb, and sports discipline. It was found that female and adolescent participants showed significant quadriceps weakness in the involved limb during the first 12 months after surgery. Female sex, younger age, and shorter time since surgery significantly increases the risk of re-injury [5,6]. Moreover the knee proprioception is persistently compromised even up to 2 years after surgery [7]. Decreased kinesthetic awareness following ACL surgery is of great importance in optimizing individual treatment plans in these patients.

Given the importance of quadriceps strength and proprioception as a predictor of the risk of recurrence, practitioners should take into account the demographic and clinical factors influencing the implementation of rehabilitation plans for decision-making processes related to return to sport. As the current literature still lacks detailed publications assessing the factors influencing the results of surgical treatment, further research is needed. 

The aim of the study was to determine the factors influencing the treatment outcome and sensory function of deep joints after applying a specific rehabilitation regimen 12 weeks after ACL electrocoagulation surgery.

## 2. Materials and Methods

The study group consisted of 41 consecutive patients with ACL tears, who underwent electrocoagulation of the femoral cruciate ligament attachment and microfracture of the femoral attachment area. The study was carried out in the period November 2019–June 2020. Inclusion criteria were age 16–45 yrs and patients operated on for partial ACL tear. Partial tear was defined as continuous fibers from the native tibial ACL footprint to the native femoral ACL footprint in arthroscopy. Exclusion criteria were combined ACL-PCL injuries, associated collateral injuries, complete ACL tear, chondral defect or bony malalignment and patients with radiographic signs of osteoarthritis. All patients suffered a knee torsion while practicing sports (amateur football, amateur basketball). The patients were treated surgically by arthroscopy of the knee joint by the same surgeon. For the purposes of this study, a questionnaire was prepared, which includes a physical and personal examination. The physical examination includes the patient’s anthropometric data, as well as information related to the cause and course of the injury. 

The test results were determined on the basis of 12 weeks of observation—the evaluation was performed 4 times: (1) before surgery, (2) on days 7–10 after surgery, (3) after 6 weeks and (4) 12 weeks after surgery. Each time, the research method was based on the evaluation of the knee stability using the front drawer test and Lachman test. 

The range of movements (extension and flexion) and muscle strength were assessed according to the Lovett scale (quadriceps muscle, muscles of the posterior group) [8]. The strength of quadriceps and hamstrings was assessed using the Lovett’ s test. The assessment was performed three times. For the quadriceps muscle, the subject was sitting in a chair. On command, he performed a knee extension movement (3 points on the Lovett scale). Then, resistance was added in the area of the proximal part of the lower leg (4 points on the Lovett scale); and for the subjects who performed the test, the resistance was determined on the distal part of the lower leg (5 points on the Lovett scale). For the hamstrings muscles, the subject was lying face down on a couch. On command, he performed the knee flexion movement (3 points on the Lovett scale). Then, resistance was added in the area of the proximal part of the lower leg (4 points on the Lovett scale); and for the subjects who performed the test, the resistance was determined on the distal part of the lower leg (5 points on the Lovett scale).

### 2.1. Surgical Electrocoagulation of the Femoral Attachment of the ACL

All surgery was carried out arthroscopically by the senior author. During surgery, the ACL injuries were assessed. In the case of partial isolated injuries or the appearance of pseudocysts, the electrocoagulation was performed—electrocoagulation of the ACL femoral attachment with bone microfractures in this area (Figure 1).

### 2.2. Assessment of Deep Feeling

The Unterberger maneuver is a dynamic equivalent test that allows us to unambiguously define the ability to react the proprioceptive sense to the changing biomechanical conditions of the patient’s body during movement. The Unterberger test is used to assess the balance, which may be severely impaired after ACL injuries. Lines determining the angular values of 0, 10, 25, and 45 degrees were marked on the ground. On a verbal command, the subject closed his eyes and walked in place, counting from 1 to 50, and then opened his eyes; the examiner, using a goniometer, assessed the position of the body in space in relation to the entry position (0 degrees) [9]. 

### 2.3. Lysholm Knee Score

Lysholm Knee Score is one of the most utilized scoring systems for ACL injuries and chondral defects [10]. The Lysholm Scale currently consists of eight items that measure: pain (25 points), instability (25 points), locking (15 points), swelling (10 points), limp (5 points), stair climbing (10 points), squatting (5 points), and need for support (5 points). Every question response has been assigned an arbitrary score on an increasing scale. The total score is the sum of each response to the eight questions and may range from 0–100. Higher scores indicate a better outcome with fewer symptoms or disabilities. 

### 2.4. Rehabilitation Procedure

Patients were equipped with a knee brace with a four-place support point and an adjustable range of movements. The change of the range of motion was systematically introduced, starting from the 2nd week after the surgery and changing the angular values of flexion by 15° in each subsequent week (for the first 2 weeks the patient had a defined range of movements in the knee joint of a value of 0–30°).

All patients after electrocoagulation of the femoral cruciate ligament attachment were subjected to rehabilitation treatment according to the procedure presented below. Each patient during the rehabilitation program was treated individually, which means that in some patients individual exercises required correction or slight modification during the exercises so that they were performed correctly—most of the subjects could easily perform the exercises in a set schedule. The same physiotherapist dealt with rehabilitation treatment for patients. The rehabilitation treatment program assumed self-exercise by patients, and the person responsible for the rehabilitation treatment process met with home-patients every 2 weeks to determine the quality of the exercises, correcting mistakes and modifying the exercises. At each control visit, the patients performed the entire prescribed set of exercises under the supervision of a physiotherapist; after each control visit, each of the subjects received written instructions to be followed. The rehabilitation treatment program lasted about 4 months, but it should be noted that the last weeks were devoted to running and jumping exercises (Table 1). 

### 2.5. Statistical Methods

The analysis used Statistica software (Version 13.0, StatSoft Inc., Tulsa, OK, USA). For dependent samples, the Wilcoxon-pair order test was used—the level of significance of differences was adopted for *p* < 0.05. However, for the independent samples the U-Mann-Whitney test was used—the significance level of the differences was set at *p* < 0.05. Spearman’s rank correlation was performed to determine the relationship between clinical parameters. The analysis was performed to evaluate variables from the independent assumption showing the mutual relationships between the analyzed factors by applying principal component analysis (PCA).

## 3. Results

### 3.1. Characteristics of the Study Group

The most numerous age group consisted of patients aged between 20–29; while the fewest patients were in the group under 20 years of age. The mean age for both women and men was 29.5 years. There were 12 women in the group of respondents, which constituted 29% of the total number of respondents; and 29 men, who respectively accounted for the remaining 71% of the studied group. Based on the analyzes carried out according to the BMI index, 18 people were of normal weight, 21 were overweight, and 1 person was slightly underweight (Table 2). 

In the group of subjects, all of the subjects had damage to the ACL: in 16 patients, the patellar articular surface was damaged; and in 12 patients, the articular cartilage was damaged in the medial femoral condyle. 

Out of 33 right-legged persons, as many as 18 of them suffered an injury to the lower limb, and 15 persons injured the non-dominant lower limb. In the group of left-footed people, the dominant side was injured in only 2 subjects, and the remaining 6 subjects were injured on the right-non-dominant side. On the basis of the performed tests, it was found that the front drawer symptom and the Lachman symptom occurred in all patients before the surgical treatment, and after the surgery, the symptoms were negative in all patients.

### 3.2. Parameters of Prioprioception at Stages of the Study

Due to the method of performing the test in the first period after the surgical treatment, the patients were not able to perform the test—in most cases it was dictated by a bad gait stereotype or concerns related to loading the lower limb, as well as body imbalance in situations when they closed their eyes. For 6 weeks, all the subjects moved with the elbow crutches; therefore, the Unterberger test was not performed in the first period, because the obtained results of the marching tests with crutches would not be reliable. On average, the group of respondents during the test performed a rotation in the range of 11–25°, and 12 patients performed the test without errors (rotation below 10°). Comparing the results before the surgery with those after the completion of the applied medical rehabilitation scheme, a strong statistically significant difference was obtained in the results of the Unterberger test in the Wilcoxon test (*p* < 0.01), which proves that the medical rehabilitation used in the program influenced the restoration of the sense of deep feeling (Table 3).

The statistically significant difference observed in the Wilcoxon test between the strength of the quadriceps muscle before and after surgery with the applied medical rehabilitation scheme (Table 3). 

Before the surgical treatment, the mean result of knee flexion for patients was 104°, while after its completion the patients improved the range of knee flexion motion to an average value of 125°. The proposed scheme of medical rehabilitation and protection of the ACL against repeated damage required the use of a knee brace with an adjustable range of motion, which meant that all patients had a 30° knee flexion range for up to 2 weeks after surgery, and for 6 weeks after surgery this value reached 90° of knee flexion. Based on the statistical analysis of the results before and after the surgery, a statistically significant difference was obtained in the parameters (Table 3). 

### 3.3. The Correlations between the Lysholm Scale and the Parameters of Deep Feeling

Based on the performed statistical analysis of the groups of women and men before and after the surgical treatment of the Lysholm scale, no statistically significant difference between the groups was noticed. Both in the group of women and men, after 12 weeks of rehabilitation, a significant improvement in the results of the Lysholm scale was observed.

The Lysholm score 12 weeks after the surgery was significantly positively correlated with the muscle strength of quadriceps (0.61 *) and hamstrings (0.48), and negatively correlated with the difference in thigh circumference (−0.44). The difference in the circumference of the lower leg did not correlate with the strength of the thigh muscles (Table 4).

The Unterberg test result after 12 weeks was significantly negatively correlated with the Lysholm score before surgery (−0.55), after 6 weeks (−0.42) and after 12 weeks (−0.42) (Table 4).

A positive correlation was found between age and the Unterberg test result at 6 (0.53) and 12 weeks (0.54) (Table 4). The Lysholm score before surgery correlated significantly with the muscle strength of the quadriceps and hamstrings at the end of the treatment and negatively with the Unterberg test at the end of the treatment. The Lysholm scale 6 weeks after surgery, negatively correlated with BMI before surgery (−0.4).

In PCA, the Lysholm scale and muscle strength were described by factor 1 independence of sex, dominant extremity and associated damage to the meniscus and cartilage (Figure 2, Figure 3, Figure 4 and Figure 5). The difference in the circumference of the thigh was described by factor 1 in men and the dominant extremity, and by factor 2 in associated damage to the meniscus and no damage to the cartilage (Figure 2, Figure 3, Figure 4 and Figure 5). The Unterberger test was described in the opposite way between men and women, and the dominant and non-dominant extremity (Figure 2 and Figure 3). 

## 4. Discussion

The study aimed to determine the impact of the effectiveness of surgical treatment along with the medical rehabilitation scheme on improving the stability and feeling of the deep knee joint and restoring the functional efficiency of the patient. We showed that the better treatment outcome was influenced by greater muscle strength and less thigh atrophy. Moreover, we found that the poorer profound sensation negatively affected the Lysholm treatment outcome. A better preoperative Lysholm score was a guarantee of greater muscle strength at the end of the observation and better deep feeling. Body weight also had a negative impact on the outcome of the therapy. The deep feeling after the therapy was that it worsened with the age of the patient. In men, we found that the asymmetry of the thigh circumference was dependent on the treatment result, while in women we did not find such a relationship. In the case of the operated dominant limb, the asymmetry of the thigh circumference influenced the treatment outcome.

One of the contentious issues of individual rehabilitation protocols is the necessity and time of using elbow crutches [11]. Most authors agree that axial loading of the lower limb is a factor determining the return to the correct gait stereotype. In most cases, the formation of a flexion contracture of the knee in the postoperative period is due to disturbances in the gait stereotype. The use of crutch support in the case of surgically treated patients does not play such an important role in relieving the lower limb as in initiating the correct gait pattern. The time of walking with elbow crutch belaying can be different and depends on many variables. The rehabilitation treatment program used at work assumed that the patients used elbow crutches for 6 weeks, i.e., until the knee joint obtains an active range of flexion to the value of 90° and correct gait stereotype is restored. Kachanathu reports that patients after ACL reconstruction should be under the protection of two elbow crutches for a period of 4 weeks [12].

In the case of procedures performed on the ligamentous apparatus of the knee joint, in most cases it is necessary to use orthoses with an adjustable range of movements to protect the ligament or graft against stretching in the first weeks. Lazaro and Dec suggest the necessity to use knee braces in the postoperative period, especially when it is necessary to reduce off-axis loads and inhibit rotational movements [13]. The authors suggest that when selecting an orthosis, the patient’s biomechanical conditions are important, and that favorable therapeutic effects can be obtained both with the use of prefabricated and custom-made orthoses, which is consistent with our own observations. Despite the fact that the use of orthoses is a kind of standard, Mayr et al. report, based on the observations made 4 years after the surgery, that there are no statistically significant differences in the group of patients without orthoses and those who used them [14]. Some studies indicate that patients led without braces reported less pain than patients with orthoses. The authors of this article conclude that the use of knee braces after ACL reconstruction using patellar tendon transplantation is not recommended. Based on our own observations of patients after the ACL femoral attachment vaporization, it can be concluded that the use of orthoses in the first period is dictated by the need to protect the ligament tense during surgery against repeated damage resulting from an excessive range of flexion.

The rehabilitation program after surgery on the knee joint is aimed at restoring the optimal range of motion in the knee joint. The nominal value of the bend in individual periods is a matter of dispute. Some authors suggest systematically increasing the range of motion in the knee joint every few days, increasing the flexion value so that the graft does not overload or the contracted ligament is stretched, while other authors suggest a more “aggressive” approach. In our study, the range of motion was increased by 30° every 2 weeks so as to achieve the full functional range of knee flexion in the period between 8 and 12 weeks. The above procedure results from the assumptions of the medical rehabilitation program. There are many rehabilitation schemes for knee injuries with similar treatment outcomes [11]. Fukuda et al. believe that working in full range of motion in the first period after surgery may result in laxity of the anterior knee compartment, which is consistent with the procedure adopted for the purposes of this study [15]. Van der List and DiFelice report that patients after reconstruction of the anatomical ligament have fewer complications than the group of patients who underwent ACL reconstruction. Moreover, patients who did not undergo reconstruction achieved the full range of movements faster, which the author of the publication explains, with the early memory of the range of movements [16]. The rehabilitation protocol allowed the patients to achieve 90° flexion in the first week and 125° one month after the surgical intervention. The procedure presented in the article of van der List and DiFelice is different than that presented in the methodology of this work; however, the resulting ranges of movements after 12 weeks are similar to each other.

Quadriceps atrophy that occurs immediately after injury is a major problem, and recovery of muscle strength and tension is a long-term process. It is estimated that it takes about 4–6 weeks to rebuild 1 cm of the thigh circumference which has disappeared due to immobilization. Most often, the circumference is lost in the range of 2–3 cm, which requires the use of improving and strengthening muscle strength for a minimum period of 3 months, which is consistent with the recommendations presented in this study and assumptions of other authors. The existing muscle imbalance or insufficient restoration of the extensor strength of the knee joint may be a predisposing factor to the occurrence of another injury. In addition, insufficient strength of the quadriceps muscle of the thigh may result in overloads within the patellofemoral joint, as well as the formation of flexion contracture. The above situations may indirectly adversely affect the biomechanics of the knee joint, which may result in changes in the mechanics of the whole body, in particular body posture, and disturbances in the gait pattern [17]. Kilgas et al. raise the need for exercises to restore the strength of the quadriceps muscle after ACL surgery [18]. Thomas et al. report that a key element in the reconstruction of the muscle circumference and its strength are strength exercises, which aim to improve the cross-section of the muscle [19]. Hauger et al. showed that percutaneous stimulation of the quadriceps muscle significantly improves its strength and efficiency in the early postoperative period, which may be a valuable guide in programming medical rehabilitation of people after surgery within the knee joint [20]. Zargi et al. observed a significant decrease in muscle strength and circumference in the first four weeks after surgery, while a significant improvement in strength to thigh volume (muscle mass) after 12 weeks, which is consistent with our observations [21]. On the basis of imaging studies, Norte et al. noticed atrophy of the quadriceps muscles of the thigh up to 20%, as well as the muscles of the posterior group of the thigh and lower leg, which may occur despite the improvement in strength and muscle contraction capacity [22]. Baron et al. reported that the intraoperative use of a tourniquet reduces the circumference of the thigh and changes the electrical signal in the EMG test, and one of the methods of rebuilding muscle mass is training with a temporary limitation of blood flow which, when the muscle performs work, is beneficial and influences it to increase its size in the cross section [23]. Lepley et al. confirmed atrophy of the quadriceps muscle of the thigh different from weight loss due to “non-use” [24].

Gait normalization is another important consideration when programming medical rehabilitation [25]. Often, disturbances in the gait pattern result from an injury and the patient moves incorrectly in the preoperative period. The factor that disturbs the gait pattern is the existing pain caused by trauma to the knee joint, hematoma and abnormal gait patterns learned in the period preceding surgery [26,27,28]. It is suggested that medical rehabilitation should be used to improve the biomechanics of the knee joint, normalize gait and prevent secondary degenerative changes in the knee joint. On the basis of EMG analyses, Gardinier et al. observed that disturbances in gait and muscle stabilization result from the reduction of peak flexion as well as the internal extensor moment of the knee joint [29]. Sigward et al. observed that the normalization of gait is expected between 8 and 12 weeks after the surgery and is a criterion enabling the introduction of plyometric and running exercises, even loading of the lower limbs [30]. Schliemann et al. showed that after surgery and medical rehabilitation, regardless of the type of surgical technique used, patients achieve the correct gait pattern [31]. Lin and Sigward note that changing the kinetics and kinematics of the knee joint contributes to the reduction of the knee extensor moments during walking, so the intervention should be aimed at improving the biomechanics of gait [32]. On the other hand, Capina et al. noticed that a comprehensive, progressive training program enabling return to sport in patients with or without disturbed gait pattern was not effective in restoring symmetry in patients 2 years after ACL reconstruction [33]. Arhos et al. noticed that the present asymmetry of gait does not only result from the asymmetry of the strength of the quadriceps muscle of the thigh, which concerned 35% of the respondents, and that restoring the strength of the quadriceps muscle alone is not enough to compensate for the asymmetry of gait [34].

The reconstruction of the anatomical continuity of the ACL ensures the mechanical stability of the knee is possible only when the individual structures of the sense of deep feeling integrate with each other. For this reason, it is recommended to use exercises that shape the sense of deep feeling already in the first stages of improvement, progressively increasing their difficulty [35]. Initially, the exercises shaping the sense of deep feeling consisted of standing one leg on a stable and then unstable ground; additionally, it is recommended that the exercising person performs exercises with eyes closed or with open eyes. In the next stages of improvement, jumping exercises, running exercises, and then with a change of pace and direction of movements are introduced. The last stage of rehabilitation may be specialized exercises consistent with the sports discipline practiced or the preferences of the patient. The assessment of deep feeling can be made by analyzing the quality of performing a given movement, as well as using specialized tests (e.g., Unterberger test, Man’s test, etc.) [36]. Patients who actively engage in sports activities, both in the amateur and professional dimension, after the rehabilitation treatment program is completed, undergo tests determining the functional efficiency of the lower limb, which was subjected to surgical treatment and improvement. Thanks to the optimization of rehabilitation programs and their individual adjustment to the needs of individual patients, the achieved treatment results are a product of many factors, and the long-term effects of treatment are emphasized by many authors [37]. Relph and Herrington suggest that there is a higher risk of secondary injuries in the group with deep sensation disorders [38,39]. Proprioceptive exercise should be part of medical rehabilitation after ACL injuries [35,40,41]. This confirms that the long-term results of the treatment of ACL lesions are largely related to the use of improvement exercises containing proprioceptive exercises, which is consistent with the results obtained by the authors. Rodriguez-Roiz et al. demonstrated that male gender, above-average age, and higher BMI had higher rates of abandoning sports activities after ACL reconstruction [42]. 

Lee et al. observed that the accelerated rehabilitation program contributes to a lower decrease in muscle strength and accelerates recovery after an injury [43]. Further comparison and verification of different programs is needed to develop a more effective medical rehabilitation program [44,45]. Grinden et al. demonstrated that the factor increasing the risk of recurrent ACL trauma is the asymmetry of strength of the quadriceps muscle after the rehabilitation program [46]. Burgi et al. showed that the guarantee of success is the time that has elapsed since the surgery, as well as the functional assessment of the patient (deep sensation and endurance tests) [47,48]. Zaffagnini et al. demonstrated that the decision to return to sport should be guided by individual reasons [49]. The above observations are consistent with our publication. One of such variables is the sense of deep feeling, which makes it possible to determine the functional state of the patient after the suffered ACL injury and the applied medical rehabilitation.

The main limitation of the research is the small size of the analyzed groups In addition to the small size of the groups, they are heterogeneous in terms of the analyzed factors. With such small, heterogeneous groups, the tools of classical statistics usually turn out to be imprecise. Therefore, taking the above into account, we decided to assess the relationship of factors using PCA. Another limitation is the lack of a control group or a reference group for the rehabilitation regimen. Our goal, however, was not to identify the factors influencing our rehabilitation program.

## 5. Conclusions

After the completion of the surgical treatment with the applied medical rehabilitation scheme, the patients regained the functional efficiency of the operated lower limb, which resulted in a feeling of stability of the knee joint and regaining the correct gait stereotype. Better treatment results were achieved in patients with greater muscle strength, less asymmetry of the thigh, better deep feeling, and younger patients with a lower body weight. The asymmetry of the muscles of the lower leg did not affect the treatment outcome. The influence of the asymmetry of the thigh circumference on the treatment outcome was related to men and in the case of the operated dominant limb.

## Figures and Tables

**Figure 1 ijerph-19-13569-f001:**
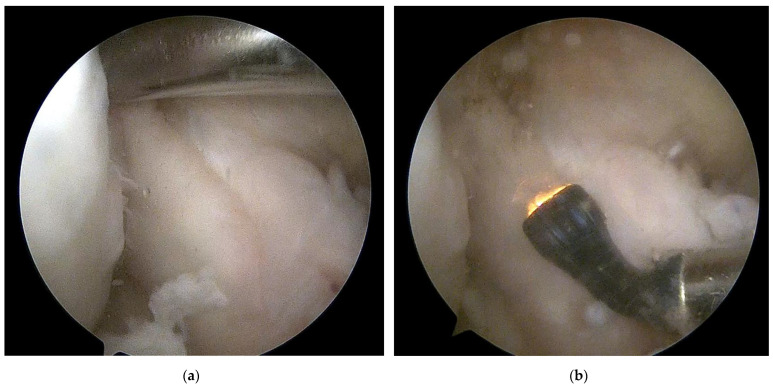
Intraoperative image of knee arthroscopy: (**a**) The method of microfracture in the area of the ACL femoral attachment (**b**) Electrocoagulation of ACL femoral attachment. Own source Pogorzała A.

**Figure 2 ijerph-19-13569-f002:**
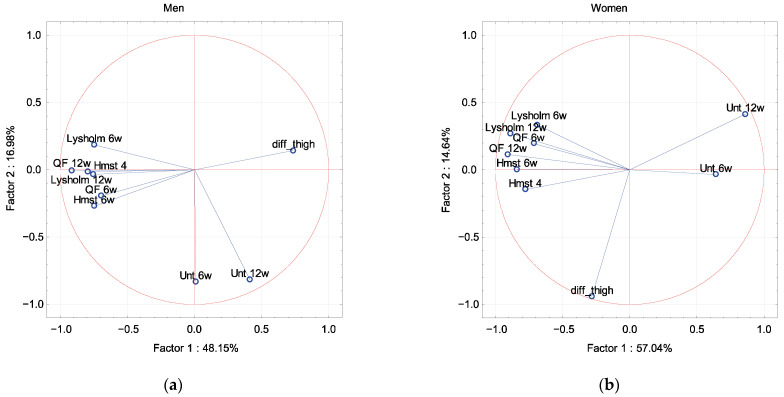
A graphic illustration of Principal Components Analysis of Lysholm, Unterberger test and muscle strength. Projection of the variables on the factor plane of the first two principal components for the men (**a**) and women (**b**).

**Figure 3 ijerph-19-13569-f003:**
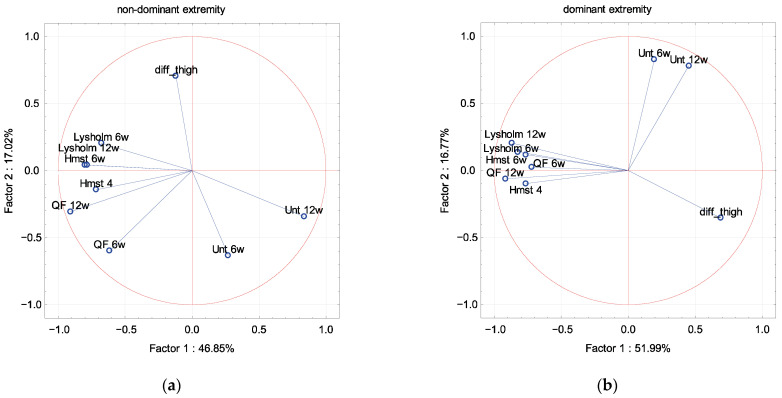
A graphic illustration of Principal Components Analysis of Lysholm, Unterberger test and muscle strength. Projection of the variables on the factor plane of the first two principal components for the non-dominant (**a**) and dominant (**b**) extremity.

**Figure 4 ijerph-19-13569-f004:**
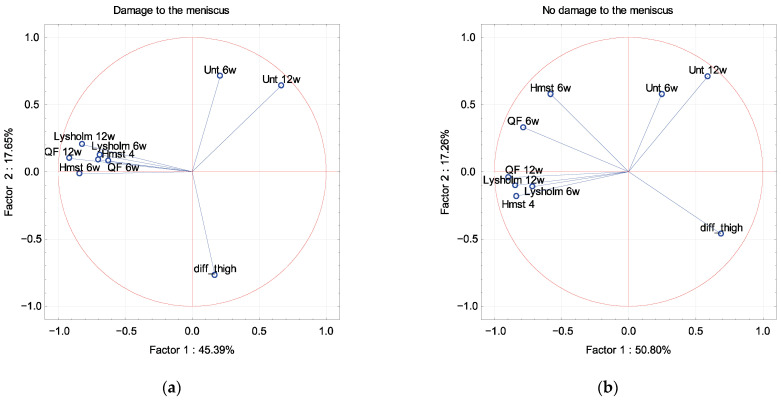
A graphic illustration of Principal Components Analysis of Lysholm, Unterberger test and muscle strength. Projection of the variables on the factor plane of the first two principal components for damage (**a**) and no damage (**b**) to the meniscus.

**Figure 5 ijerph-19-13569-f005:**
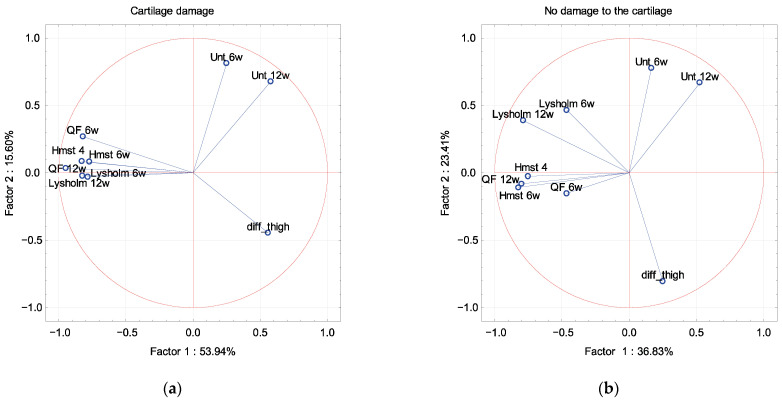
A graphic illustration of Principal Components Analysis of Lysholm, Unterberger test and muscle strength. Projection of the variables on the factor plane of the first two principal components for damage (**a**) and no damage (**b**) to the cartilage.

**Table 1 ijerph-19-13569-t001:** Rehabilitation treatment protocol.

Phase	Aim	Frequency of Exercises	Sample Exercises
I early (0–10 days after surgery)	preventing swelling, restoration of the extension, learning to move correctly in the belaying of elbow crutches, protection of the ligament against stretching	every 1 h for 10 repetitions	Anticoagulant exercise Isometric exercises Extending the limb up (SLR)
II restoration of muscle strength and range of movements (11 days–6 weeks)	preventing swelling, extension normalization, learning to move correctly in the belaying of elbow crutches on flat surfaces and stairs, protection of the ligament against stretching, restoration of the strength of the thigh muscles, restoration of the range of movements up to 90 degrees of flexion, knee brace	The whole set of exercises 2 times a day	Knee bending while sliding the heel on the ground (each week the range of bending in the knee joint should be increased by 15 degrees: SLR Exercises to strengthen adductors Learning the axial loading of the operated lower limbs Ex. proprioception in the supine and standing position Strengthening the strength of the muscles of the anterior and posterior thigh groups (pulling the tape) SLR seated on a chair with rubber March in place Strengthen the muscles of the posterior thigh group Stretching
III rebuilding deep feeling (7–12 weeks-day)	preventing swelling, improving movement in the correct gait pattern on flat surfaces and stairs, protection of the ligament against stretching, restoration of the strength of the thigh muscles, restoration of the full range of movements, knee brace for walking outside the house, rebuilding deep feeling	The whole set of exercises 2 times a day	Recreate the full range of movements SLR in different starting positions Half-squats in a closed biomechanical chain Proprioception exercises on an unstable surface Exercises with eyes open and closed Marches on flat surfaces and stairs Stretching
IV plyometric exercises/improvement exercises (13–16 weeks)	enhancing deep feeling, running and jumping exercises	The whole set of exercises 1 time a day	SLR in various positions Marching on the stairs Running and jumping Jumping one-legged Multi-jumps Marches with a change of pace and direction of movements Exercises in open biomechanical chains Proprioception exercises on an unstable surface Special exercises in selected sports disciplines Stretching

**Table 2 ijerph-19-13569-t002:** Characteristics of the study group.

	Women (*n* = 12)/Men (*n* = 29)	All
Age (mean ± SD)	29.5 ± 8.9/29.5 ± 8.1	29.5 ± 8.3
Body mass (mean ± SD)	62.2 ± 7.6/80.6 ± 7.3	75.2 ± 11.2
Height (mean ± SD)	170.5 ± 4.9/176.8 ± 5.2	174.9 ± 5.9
BMI (mean ± SD)	21.4 ± 2.2/25.8 ± 1.8	24.5 ± 2.8
Dominant surgical knee (%)	14 (34.1%)/6 (14.6%)	20 (48.8%)
Thigh circumference difference	1.8 ± 0.6/1.8 ± 0.6	1.8 ± 0.6
Shin circumference difference	1.2 ± 0.6/1.1 ± 0.6	1.1 ± 0.6

**Table 3 ijerph-19-13569-t003:** Mean values with standard deviation for the Unterberg test, muscle strength test and Lysholm scale at individual stages of observation. QF—quadriceps muscle, Hmstr—muscle strength of the posterior group.

Stage	Unterberg	Strength QF	Strength Hmstr	Flexion	Lysholm Scale
before	-	3.9 ± 0.3	4 ± 0.1	104.4 ± 11.2	44.5 ± 16.4
7–10 days	-	3.8 ± 0.3	4 ± 0.1		64.3 ± 4.6
6 weeks	40.7 ± 3.6	4.4 ± 0.3	4.4 ± 0.2		84.3 ± 5.9
12 weeks	16.7 ± 7.3	4.8 ± 0.3	4.8 ± 0.3	121.2 ± 7.8	98.9 ± 1.7

**Table 4 ijerph-19-13569-t004:** Values of correlation coefficients between the Lysholm scale and the parameters of deep feeling, muscle strength, BMI and age.

		Lysholm Scale				Age	BMI	Circumf. Diff.	
	Time Surgery	Before	7–10 Days	6 Weeks	12 Weeks			Shank	Thigh
Unterberger	6 weeks	−0.24	0.10	−0.20	−0.09	0.53 *	0.11	0.15	−0.04
	12 weeks	−0.55 *	−0.06	−0.42 *	−0.42 *	0.54 *	0.22	0.05	−0.15
quadriceps muscle strength	before	0.47 *	0.13	0.41 *	0.33 *	−0.07	0.12	−0.20	−0.11
	7–10 days	0.20	0.01	0.21	0.24	−0.04	0.17	−0.21	−0.21
	6 weeks	0.41 *	0.16	0.30	0.41 *	0.07	0.02	−0.26	−0.35 *
	12 weeks	0.49 *	0.30	0.49 *	0.61 *	−0.12	−0.08	−0.11	−0.35 *
Hamstrings’ muscle strength	before	0.22	0.21	0.09	0.05	0.18	0.11	−0.26	−0.07
	7–10 days	−0.05	−0.21	0.15	0.19	−0.14	−0.22	−0.06	−0.31 *
	6 weeks	0.40 *	0.28	0.34 *	0.56 *	−0.01	−0.20	−0.16	−0.21
	12 weeks	0.50 *	0.38 *	0.51 *	0.48 *	−0.05	−0.11	−0.10	−0.21
age		−0.15	0.21	−0.12	−0.20				
BMI		−0.08	−0.07	−0.40 *	−0.30	0.21			
circumf. diff.	shank	−0.20	−0.11	−0.24	−0.12	0.08	0.19		
	thigh	−0.03	−0.02	−0.32 *	−0.44 *	−0.28	0.39 *	0.24	

* Statistically significant.

## Data Availability

Data available on request.

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
