# Peer review of "Factors Influencing Treatment Outcome and Proprioception after Electrocoagulation of the Femoral Insertion of the Anterior Cruciate Ligament"

_ijerph, 2022, doi:10.3390/ijerph192013569_

Round 1

Reviewer 1 Report

The study makes important contributions to the field of orthopedic rehabilitation.

-The introductory part of the study is well written.

-In the method section, it should be stated where the study was carried out, between which dates and in which sports branch the participants were in.

-However, the inclusion and exclusion criteria of the participants should be given in detail.

-In addition, information should be given about how the sample size of the study was calculated, why were 41 people included?

-The Results and Discussion part of the study are clearly written.

Author Response

Response to Reviewer 1 Comments

We would like to express our sincere appreciation of the reviewers’ detailed examination of our paper and the resulting comments and suggestions. Many good points have been raised, which have helped us present our study more clearly. The manuscript has been revised based on these suggestions; all changes are highlighted in the revised version of the manuscript. Our point-by-point responses to the reviewers’ comments are given below.  

Point 1 

The introductory part of the study is well written.

Thank you for this comment. 

Point 2 

In the method section, it should be stated where the study was carried out, between which dates and in which sports branch the participants were in.

Thank you for this comment. Accordingly, we have added information in section Materials and Methods.

Point 3 -However, the inclusion and exclusion criteria of the participants should be given in detail.

Thank you for this comment. Accordingly, we have added  the criteria in section Materials and Methods.

Point 4 -In addition, information should be given about how the sample size of the study was calculated, why were 41 people included?

Thank you for this comment. The means 41 are needed to have a confidence level of 80% that the real value is within ±10% of the measured value.

Point 5 -The Results and Discussion part of the study are clearly written.

Thank you for this comment.

Reviewer 2 Report

In the manuscript, the author mainly evaluated the factors that could affect the treatment outcome and deep feeling function following a specific rehabilitation scheme for the patients with anterior cruciate ligament vaporization surgery. Their results indicated that greater muscle strength, less thigh atrophy, better deep sensation, and younger patients with a lower body weight could achieve a better treatment results. Overall, the study is meaningful, and the manuscript is well organized. However, several issues are required to be addressed and clarified in the study.

Abstract

The results is vague. There was a statistically significant improvement in most of the parameters assessed. Please list the parameters. And please clarify main finding in the results.

Moreover, the conclusion is the same as the last section of the manuscript. Please rephrase it. 

Introduction

Please clarify the rationale for the study. Why did the author would like to determine the factors influencing the treatment outcome? What is unclear for the medical rehabilitation methods in the patients with ACL surgery? 

Methods

For the patient recruitment, please clarify the criteria in the details. Are the patients allowed to take any medications in the study?  Some medications may affect the score for the pain and swelling. 

In addition, for the section of characteristics of the study group in the results, the author may consider to put it the methods section. 

Results

For the sub-section, please list 3.2 and 3.3 with sub-title to make the reader easily follow and understand. 

Author Response

Response to Reviewer 2 Comments

We would like to express our sincere appreciation of the reviewers’ detailed examination of our paper and the resulting comments and suggestions. Many good points have been raised, which have helped us present our study more clearly. The manuscript has been revised based on these suggestions; all changes are highlighted in the revised version of the manuscript. Our point-by-point responses to the reviewers’ comments are given below. 

Point 1 

Abstract. The results is vague.  There was a statistically significant improvement in most of the parameters assessed. Please list the parameters. And please clarify main finding in the results.

Thank you for this comment. Accordingly, we have changed the abstract. 

Point 2 

Abstract. 

Moreover, the conclusion is the same as the last section of the manuscript. Please rephrase it. 

Thank you for this comment. Accordingly, we have changed the abstract. 

Point 3 

Introduction

Please clarify the rationale for the study. 

Why did the author would like to determine the factors influencing the treatment outcome? 

What is unclear for the medical rehabilitation methods in the patients with ACL surgery? 

Thank you for this comment. Accordingly, we have added  the rationale for the study.

Point 6

Methods 

For the patient recruitment, please clarify the criteria in the details. 

Thank you for this comment. Accordingly, we have added  the criteria in section Materials and Methods.

Point 7

Methods 

Are the patients allowed to take any medications in the study? Some medications may affect the score for the pain and swelling. 

Thank you for this comment.  The patients received painkillers for up to 2 weeks after the procedure (NSAID and tramadol).

Point 8

Methods 

In addition, for the section of characteristics of the study group in the results, the author may consider to put it the methods section. 

Thank you for this comment.  Accordingly, we have changed.

Point 9
Results 

For the sub-section, please list 3.2 and 3.3 with sub-title to make the reader easily follow and understand. 

Thank you for this comment. Accordingly, we have added  subtitles.

Reviewer 3 Report

I believe the paper suffers some major flaws. First of all, no information is provided on the “vaporization of the femoral cruciate ligament attachment”. The surgical approach is not mentioned nor described in the introduction, it is not described in the methods, and no bibliography is provided on the topic. Searching “ACL vaporization surgery” on PubMed (or on Google) doesn’t seem to provide results. More generally speaking, the introduction delivers only generic information on ACL lesions and treatments, and it doesn’t “build up” the information needed to introduce the topic of the paper. Moreover, it is not clear what the inclusion and exclusion criteria of the study were. References are missing for many of the tests used (e.g., Lovett scale, Unterberger maneuver), and bibliography is needed to support statements such as “[it] allows us to unambiguously define the ability to react the proprioreceptive sense to the changing biomechanical conditions of the patient's body during movement”, in lines 72-74. Also, some parts seem to be poorly written and never checked before submission (e.g., “wypełniano na każdym etapie” in line 79).

I’m sure the authors will be able to manage these problems and submit a new version of this article after a substantial re-writing has been performed.

Author Response

Response to Reviewer 3 Comments

We would like to express our sincere appreciation of the reviewers’ detailed examination of our paper and the resulting comments and suggestions. Many good points have been raised, which have helped us present our study more clearly. The manuscript has been revised based on these suggestions; all changes are highlighted in the revised version of the manuscript. Our point-by-point responses to the reviewers’ comments are given below.   

Point 1

First of all, no information is provided on the “vaporization of the femoral cruciate ligament attachment”. The surgical approach is not mentioned nor described in the introduction, it is not described in the methods, and no bibliography is provided on the topic. Searching “ACL vaporization surgery” on PubMed (or on Google) doesn’t seem to provide results. 

Thank you for this comment. Accordingly, we have added  in the section Introduction and  in section Materials and Methods.

Point 2

More generally speaking, the introduction delivers only generic information on ACL lesions and treatments, and it doesn’t “build up” the information needed to introduce the topic of the paper. 

Thank you for this comment. Accordingly, we have added  in the section Introduction.

Point 3

Moreover, it is not clear what the inclusion and exclusion criteria of the study were. 

Thank you for this comment. Accordingly, we have added  the criteria in section Materials and Methods.

Point 4

References are missing for many of the tests used (e.g., Lovett scale, Unterberger maneuver), and bibliography is needed to support statements such as “[it] allows us to unambiguously define the ability to react the proprioreceptive sense to the changing biomechanical conditions of the patient's body during movement”, in lines 72-74. 

Thank you for this comment. Accordingly, we have added  the references.

Point 5

Also, some parts seem to be poorly written and never checked before submission (e.g., “wypełniano na każdym etapie” in line 79).

 Thank you for this comment. Accordingly, we have corrected inconsistencies.

Round 2

Reviewer 2 Report

Thank you for the revision. The author addressed my concerns in the revised manuscript. 

Author Response

Response to Reviewer 2 Comments

Thank you for the revision. The author addressed my concerns in the revised manuscript. 

Thank you.

Reviewer 3 Report

I have read the revised version of the article, finding that the introduction section has been remarkably improved. However, I still have many observations and I believe the paper needs major revisions.

In details:

·        It is unclear how strength (quadriceps and hamstrings) was assessed.

·        It is unclear how the statistical significance of the correlations was determined.

·        No correction for multiple comparisons (i.e., Bonferroni) was applied. Such correction should be used when multiple measures and outcome indexes are involved, like in the present study.

·        It is unclear what led to the use of the Unterberger test in ACL-injured patients.

·        About “vaporization surgery”, the reference you provided (Lamar et al.) never uses the term “vaporization”. It says that “The ACL underwent modification until blanching and tightening were accomplished”.

Also, the English language should be seriously revised throughout the whole manuscript, as it is not acceptable in the current version of the paper (some examples are provided further).

Some minor revisions:

Lines 137-138: “During surgery assessed the ACL injuries” should be, I guess, “During surgery, the ACL injuries were assessed”.

Lines 138-143: “In the case partial isolated injuries or the appearance of pseudocysts performed the electrocoagulation - vaporization of the ACL femoral attachment with bone microfractures in this area (Fig. 1).”. English language should be revised.

Lines 147-152: what do you mean by “Rotation above 45 ° made the test impossible”? I didn’t find such information in the reference you provided.

Lines 154-155: “Lysholm Knee Score is one of the most utilized scoring systems for ACL injuries and chondral defects was completed at each stage”. English language should be revised.

Lines 169-170: “Particular periods of introducing exercises were normally adjusted for individual patients”. The sentence is not clear.

Line 186: “which can only be started after restoring the sense of deep feeling.”. References are needed for such statement.

Table 1: “Streatching” is incorrect.

Lines 205-206: “in 24 patients (59%)”. It is unclear. The sentence looks incomplete.

Characteristics of the study group should be part of the Results section.

Section 3.1: “Due to the method of performing the test in the first period after the surgical treatment, the patients were not able to perform the test […] After 6 weeks from the commencement of rehabilitation, a significant improvement was noticed”. It is not clear. How could you detect the improvement, when the tests were not performed at the beginning of rehabilitation?

Lines 249-250: “which meant that all patients had a 30 ° knee flexion range for up to 2 weeks after surgery, and for 6 weeks after surgery”. It is not clear. Two weeks or six weeks?

Line 250: “After hour”. What does it mean?

Line 399: “percutaneous stimulation of the quadriceps muscle”. What kind of stimulation?

Author Response

Response to Reviewer 3 Comments

We would like to express our sincere appreciation of the reviewers’ detailed examination of our paper and the resulting comments and suggestions. Many good points have been raised, which have helped us present our study more clearly. The manuscript has been revised based on these suggestions; all changes are highlighted in the revised version of the manuscript. Our point-by-point responses to the reviewers’ comments are given below.   

I have read the revised version of the article, finding that the introduction section has been remarkably improved. However, I still have many observations and I believe the paper needs major revisions.

Point 1

  •       It is unclear how strength (quadriceps and hamstrings) was assessed.

Thank you for this comment.  Accordingly, we have changed.

The strength of quadriceps and hamstrings was assessed using Lovett’ s test.

The study was performed three times. For the quadriceps muscle, the subject was sitting in a chair. On command, he performed a knee extension movement (3 points on the Lovett scale). Then, resistance was added in the area of the proximal part of the lower leg (4 points on the Lovett scale), and for the subjects who performed the test, the resistance was determined on the distal part of the lower leg (5 points on the Lovett scale).

For the hamstrings muscles, the subject was lying face down on a couch. On command, he performed the knee flexion movement (3 points on the Lovett scale). Then, resistance was added in the area of the proximal part of the lower leg (4 points on the Lovett scale), and for the subjects who performed the test, the resistance was determined on the distal part of the lower leg (5 points on the Lovett scale).

Point 2

  •       It is unclear how the statistical significance of the correlations was determined.

For dependent samples, the Wilcoxon-pair order test was used - the level of significance of differences was adopted for p <0.05. However, for the independent samples the U-Mann-Whitney test was used – the significance level of the differences was set at p <0.05. 

Point 3

  •       No correction for multiple comparisons (i.e., Bonferroni) was applied. Such correction should be used when multiple measures and outcome indexes are involved, like in the present study.

Lysholm scale

age

BMI

circumf. diff.

Time

surgery

before

7-10 days

6 weeks

12 weeks

shank

thigh

Unterberger

6 weeks

-0.24

0.10

-0.20

-0.09

0.53* (0,0004)

0.11

0.15

-0.04

12 weeks

-0.55* (0,0002)

-0.06

-0.42*
(0,006)

-0.42*
(0,006)

0.54*
(0,0002)

0.22

0.05

-0.15

quadriceps muscle strength

before

0.47*
(0,001)

0.13

0.41*
(0,007)

0.33*
(0,036)

-0.07

0.12

-0.20

-0.11

7-10 days

0.20

0.01

0.21

0.24

-0.04

0.17

-0.21

-0.21

6 weeks

0.41*
(0,007)

0.16

0.30

0.41*
(0,007)

0.07

0.02

-0.26

-0.35* (0,02)

12 weeks

0.49* (0,001)

0.30

0.49*

0.61*
(0,00002)

-0.12

-0.08

-0.11

-0.35* (0,02)

Hamstrings' muscle

strength

before

0.22

0.21

0.09

0.05

0.18

0.11

-0.26

-0.07

7-10 days

-0.05

-0.21

0.15

0.19

-0.14

-0.22

-0.06

-0.31*
(0,04)

6 weeks.

0.40*
(0,01)

0.28

0.34*
(0,02)

0.56*
(0,0001)

-0.01

-0.20

-0.16

-0.21

12 weeks

0.50*
(0,0009)

0.38*

0.51*
(0,0006)

0.48*
(0,001)

-0.05

-0.11

-0.10

-0.21

age

-0.15

0.21

-0.12

-0.20

BMI

-0.08

-0.07

-0.40*
(0,009)

-0.30

0.21

circumf. diff.

shank

-0.20

-0.11

-0.24

-0.12

0.08

0.19

thigh

-0.03

-0.02

-0.32*

-0.44*
(0,003)

-0.28

0.39*
(0,01)

0.24

We have included the p values for the obtained correlation coefficients in the table. They are significantly lower than 0.01, therefore it can be concluded that the Bonferroni correction (dividing alpha by even 30) will maintain statistical significance. 

Point 4

  •       It is unclear what led to the use of the Unterberger test in ACL-injured patients.

Thank you for this comment.  Accordingly, we have changed.

The Unterberger test is used to assess the balance that may be severely impaired after ACL injuries.· Lines determining the angular values of 0, 10, 25, 45 degrees were marked on the ground. On a verbal command, the subject closed his eyes and walked in place, counting from 1 to 50, and then opened his eyes,  the examiner, using a goniometer, assessed the position of the body in space in relation to the entry position (0 degrees).

Point 5

  •       About “vaporization surgery”, the reference you provided (Lamar et al.) never uses the term “vaporization”. It says that “The ACL underwent modification until blanching and tightening were accomplished”.

In fact, vaporization is quite a common word in arthroscopic surgery. A proposal to switch to  electrocoagulation.

Point 6

Lines 137-138: “During surgery assessed the ACL injuries” should be, I guess, “During surgery, the ACL injuries were assessed”.

Thank you for this comment.  Accordingly, we have changed.

Point 7

Lines 138-143: “In the case partial isolated injuries or the appearance of pseudocysts performed the electrocoagulation - vaporization of the ACL femoral attachment with bone microfractures in this area (Fig. 1).”. English language should be revised.

Thank you for this comment.  Accordingly, we have changed.

Point 8

Lines 147-152: what o you mean by “Rotation above 45 ° made the test impossible”? I didn’t find such information in the reference you provided.

Thank you for this comment.  Accordingly, we have changed.

Point 9

Lines 154-155: “Lysholm Knee Score is one of the most utilized scoring systems for ACL injuries and chondral defects was completed at each stage”. English language should be revised.

Thank you for this comment.  Accordingly, we have changed.

Point 10

Lines 169-170: “Particular periods of introducing exercises were normally adjusted for individual patients”. The sentence is not clear.

Thank you for this comment. Each patient during the rehabilitation program was treated individually, which means that in some patients individual exercises required correction or slight modification during the exercises so that they were performed correctly 

Point 11

Line 186: “which can only be started after restoring the sense of deep feeling.”. References are needed for such statement.

Thank you for this comment.  Accordingly, we have changed.

Point 12

Table 1: “Streatching” is incorrect.

Thank you for this comment.  Accordingly, we have changed.

Point 13

Lines 205-206: “in 24 patients (59%)”. It is unclear. The sentence looks incomplete.

Thank you for this comment.  Accordingly, the text fragment we have removed.

Point 14

Characteristics of the study group should be part of the Results section.

Thank you for this comment.  Accordingly, we have changed.

Point 15

Section 3.1: “Due to the method of performing the test in the first period after the surgical treatment, the patients were not able to perform the test […] After 6 weeks from the commencement of rehabilitation, a significant improvement was noticed”. It is not clear. How could you detect the improvement, when the tests were not performed at the beginning of rehabilitation?

Thank you for this comment. For 6 weeks, all the subjects moved with the elbow crutches, therefore the Unterberger test was not performed in the first period, because the obtained results of the marching tests with crutches would not be reliable.

Point 16

Lines 249-250: “which meant that all patients had a 30 ° knee flexion range for up to 2 weeks after surgery, and for 6 weeks after surgery”. It is not clear. Two weeks or six weeks?

Thank you for this comment. Every 2 weeks the flexion was increased by 30 degrees; in the 6th week a 90-degree bend was achieved

Point 17

Line 250: “After hour”. What does it mean?

Thank you for this comment.  Accordingly, we have corrected it.